# Multi-drug resistant bacteria predict mortality in bloodstream infection in a tertiary setting in Tanzania

**Joel Manyahi**[1]*, **Upendo Kibwana**[1], **Edna Mgimba**[2], **Mtebe Majigo**[1]

**1** Department of Microbiology and Immunology, Muhimbili University of Health and Allied Sciences, Dar es Salaam, Tanzania, **2** Central pathology laboratory, Muhimbili National Hospital, Dar es Salaam, Tanzania

* manyahijoel@yahoo.com

**Data Availability Statement:** All relevant data are within the paper and its Supporting Information file.

**Funding:** The author (s) received no specific funding for this work.

## Abstract

### Background

Bloodstream infections (BSI) are serious and life-threatening, associated with high mortality and morbidity. In resource-limited settings, there is a paucity of data on predictors of outcome in patients with BSI. This study aimed at examining the predictors of mortality in patients with BSI as well as bacteria causing BSI.

### Methods and materials

This was a cross-sectional study conducted at Muhimbili National Hospital between April and May 2018. Blood culture results from all inpatients at the clinical microbiology laboratory were recorded and clinical information was retrieved retrospectively from the files. Bacteria from positive blood culture were identified and antimicrobial susceptibility was performed.

### Results

The overall prevalence of BSI was, 46/402 (11.4% 95% CI 8.6–15), with a case fatality rate of 37%. There was a significantly high rate of BSI in patients who had died (19.5%) compared to those who survived (9.2%) p = 0.008. Gram-negative bacteria (74%) were the common cause of BSI, with a predominance of Enterobacteriaceae (22), followed by *Pseudomonas aeruginosa* (11). The majority of bacteria (70.5%) isolated from patients with BSI were Multi-drug resistant (MDR). Forty-six percent of *Pseudomonas aeruginosa* were resistant to meropenem while 68% (15/22) of Enterobacteriaceae were extended-spectrum β lactamase producers. Carbapenemase production was detected in 27% (3/11) of *Pseudomonas aeruginosa* and one *Proteus mirabilis*. Forty percent of *Staphylococcus aureus* were methicillin-resistant *Staphylococcus aureus*. Positive blood culture (aOR 2.24, 95%CI 1.12–4.47, p 0.02) and admission to the intensive care unit (aOR 3.88, 95%CI 1.60–9.41, p = 0.003) were independent factors for mortality in suspected BSI. Isolation of MDR bacteria was an independent predictor for mortality in confirmed BSI (aOR 15.62, 95%CI 1.24–161.38, p = 0.02).

**Competing interests:** No authors have competing interests.

## Conclusion

The prevalence of BSI was 11.4%, with the majority of bacteria in BSI were MDR. Positive blood culture, admission to the ICU and MDR were predictors for mortality.

## Introduction

Bloodstream infection (BSI) is life-threatening not only associated with increased mortality, and morbidity but also health care costs [1]. Often, MDR bacteria causing BSI are associated with poor patient outcome compared to susceptible bacteria [2, 3]. The incidence and prevalence of BSI vary considerably between developed and developing countries [4–6]. Nevertheless, the epidemiology of BSI in both community and hospital settings is evolving. Besides, data on the rapid changing bacterial etiology of BSI in Tanzania is scarce.

Treatment of BSI in the resource-limited setting is largely empirical using broad-spectrum antibiotics. Empiric treatments often fail to target the correct pathogens, leading to treatment failures and increasing mortality [1, 7]. To address these, clinical microbiology laboratories may play important roles in the effective management of BSI. Prompt reporting of results coupled with identifying critical values and antibiogram pattern provided by laboratories facilitate the successful management of patients with BSI. Elsewhere, factors predicting mortality in BSI have been investigated [2, 8] but few data exist in Tanzania.

The increasing burden of hospital-acquired BSI caused by Multidrug-resistant (MDR) pathogens including extended-spectrum beta-lactamases (ESBL) have been previously observed in Tanzania [3]. However, in a world of rapidly evolving bacteria pathogens, repeated surveillance is warranted for improving the management of BSI. Our study aimed at identifying the current bacterial etiology and predictors of mortality in BSI.

## Materials and methods

### Study design and setting

This was a cross-sectional study conducted at Muhimbili National Hospital (MNH) between April and May 2018. MNH is the largest tertiary hospital in Tanzania, serving approximately 6 million people from Dar es Salaam. It has a 1500 bed capacity, attending approximately 1200 inpatients per week and approximately 1200 outpatients per day. MNH is also a training facility for the Muhimbili University of Health and Allied Sciences and the main referral hospital in Tanzania.

### Study population

The study included all inpatients with clinical features suggestive of BSI, from whom blood specimens for culture were processed at MNH Clinical Microbiology Laboratory.

### Data collection

A structured data collection tool was used to record results of blood culture, colonial morphology, Gram stain, isolates identity and antimicrobial susceptibility test (AST). Demographic characteristics such as sex, age, and other patients' information were extracted from patient request forms. Patients' clinical outcomes were retrieved from the patient's clinical case notes.

## Blood culture and bacterial identification

Blood was collected by the attending clinician into blood culture bottles for adult (BD BAC-TEC Plus Aerobic /F Culture Vials, Becton Dickinson and Company) and pediatric (BD BAC-TEC Peds Plus™/F Culture Vials, Becton Dickinson and Company) and used for culture. Upon reaching the laboratory, blood culture bottles were inspected for acceptance criteria. Blood culture vials were incubated into the BD BACTEC FX40 analyzer for a maximum of five days.

Primary Gram stain was performed on positive cultures followed by subculture on appropriate solid culture media. A single drop of blood was inoculated into 5% sheep blood agar (SBA) and MacConkey agar (MCA), then incubated at 37˚C with 5–10% $CO_2$ and 37˚C respectively for 18–24 hours. Bacteria were initially identified by colony morphology and Gram stain. Gram-positive cocci were further identified by a set of biochemical tests including, catalase test, coagulase, DNase, Staphaurex (Remel Europe Ltd, Dartford, UK), *Streptococcus* grouping kit (Remel Europe Ltd, Dartford, UK). Gram-negative rods were further identified by API20 E and API20 NE (Biomerieux, France).

## Antimicrobial susceptibility testing

Kirby Bauer disc diffusion method was used to test antimicrobial susceptibility following Clinical and Laboratory Institute guidelines [CLSI] [9]. Antibiotics included in susceptibility testing were those commonly used in our settings for the management of BSI and few reserved for severe bacterial infections.

MDR was defined as resistance to at least one antibiotic in three or more antimicrobial classes [10]. Methicillin Resistant *Staphylococcus aureus* (MRSA) was determined by the disc diffusion method using cefoxitin disk (30μg) (Oxoid, UK) as previously described [9]. ESBL production in Enterobacteriaceae was detected as described previously [9]. Carbapenemase production in Gram-negative bacteria was screened using a combination disk method whereby meropenem disk (10μg) alone and a combination of meropenem disk (10μg) with dipicolinic acid (DPA) 1000μg. An increase of zone diameter for 5 mm or more around combined meropenem disk and DPA, as compared with the disk of meropenem alone, was considered to be a positive result [11].

The following reference strains were used for quality control: *Escherichia coli* (*E. coli*) ATCC 25922, *Klebsiella pneumoniae* ATCC 700603 for ESBL, *Staphylococcus aureus* ATCC 25923, and *Staphylococcus aureus* ATCC 29213 for MRSA and *Klebsiella pneumoniae* ATCC 1705 and *Klebsiella pneumoniae* ATCC 1706 for carbapenemase resistance.

## Data analysis

Statistical analysis was performed using SPSS Version 25.0 (Armonk, NY: IBM Corp). Descriptive analysis for categorical variables was summarized in the form of frequencies and percentages. The comparison within variables was performed using the Chi-square test or Fisher´s exact test to observe the proportion differences. Binary univariate regression analysis was performed to identify factors associated with mortality. All variables with $p < 0.2$ at univariate analysis were further analyzed in multivariate binary regression to identify independent factors associated with mortality. P-value $< 0.05$ was considered statistically significant.

## Ethical approval

Ethical approval for the study was obtained from the Senate Research and Publication Committee, the Institutional Review Board of the Muhimbili University of Health and Allied

Science. Permission to conduct the study at Muhimbili National Hospital was granted by executive director. Informed consent was waived by ethical committee and Muhimbili National Hospital, because the study was retrospective and they were no direct contact with patients.

## Results

### Description of the study participants

A total of 402 specimens for blood cultures were included in the study. The majority (39.3%) were obtained from patients aging 1–14 years and 235 (58.5%) were from males. Most blood cultures (40.8%) were from pediatric wards while 16.9% (68/402) were from patients admitted in intensive care unit (ICU). More than half (57.7%) had a history of antibiotic use prior to blood culture. Ceftriaxone was the most commonly used antibiotic, followed by meropenem. A total of 87(21.6%) patients died during the study period (Table 1).

**Table 1. Demographic and clinical characteristics of study participants.**

| Variable | Frequency (%) |
|---|---|
| **Age** | |
| <1 month | 53 (13.2) |
| 1–14 years | 158 (39.3) |
| 15–46 | 113 (28.1) |
| >46 | 78 (19.4) |
| **Sex** | |
| Male | 235 (58.5) |
| Female | 167 (41.5) |
| **Ward** | |
| Pediatric | 164 (40.8) |
| Surgical | 79 (19.7) |
| Medical | 91 (22.6) |
| ICU | 68 (16.9) |
| **Antibiotic use before culture** | |
| Yes | 232 (57.7) |
| No | 170 (42.3) |
| **Antibiotics administered before culture** | |
| Ceftriaxone | 127 (31.6) |
| Ampicillin/Gentamicin | 21 (5.2) |
| Amoxycillin-clavulanic acid | 21 (5.2) |
| Meropenem | 31 (7.7) |
| Ciprofloxacin/Vancomycin | 18 (4.5) |
| Others | 14 (3.4) |
| No | 170 (42.3) |
| **Underlying diseases** | |
| Malignant | 93 (23.1) |
| Kidney diseases | 55 (13.7) |
| Sepsis | 131 (32.6) |
| Stroke/post-surgery/Diabetes mellitus | 76 (18.9) |
| Others | 47 (11.7) |
| **Outcome** | |
| Death | 87 (21.8) |
| Discharged | 315 (78.1) |

## Rate of laboratory confirmed bloodstream infections

There were 402 participants with blood culture included in the study, 11.4% (46/402) (95% CI 8.6–15) had culture-positive BSI, and 17 of these patients died (case fatality rate was 37%). The rate of culture-positive BSI was slightly higher in females (12%, 20/167) than in males (11.1%, (26/235). There was no statistically significant difference in the rate of BSI in the neonates (15.1%, 8/53), children aged 1–14 years (7.6%, 12/158), patients aged 15–46 years (13.3%, 15/113) and patients aged more than 46 years (14.1%, 11/78), $p$ = 0.3. Patients admitted to the ICU had a higher incidence of BSI (17.6%, 12/68) compared to those admitted to the pediatric (8.5%, 14/164), surgical (8.9%, 7/79) and medical ward (14.3%, 13/91) p = 0.2. There was a significantly high rate of culture-positive BSI in deceased patients (19.5%) compared to those that survived (9.2%), p = 0.008.

## Bacterial isolates and antimicrobial susceptibility pattern

A total of 46 pathogens were isolated from blood cultures. The majority (74%) were Gram-negative bacteria, of which *Klebsiella pneumoniae* and *Pseudomonas aeruginosa* were the most common accounting for 23.9% each. *Acinetobacter baumannii* and *Salmonella* Typhi accounted for one isolate each. *Staphylococcus aureus* was only Gram-positive bacteria isolated accounting for 10 isolates and *Candida albicans* contributed 2 isolates. (Fig 1). Antimicrobial susceptibility test was performed for 43 bacterial isolates to determine susceptibility patterns. The majority of isolates (70.5%, 31/44) were MDR. We found patients who used antibiotics prior to blood culture were at increased risk for isolation of MDR bacteria (OR 4.86, 95% CI 1.14–20.70, p = 0.03) compared to those who did not use antibiotics (Table 2)

Sixty-eight percent (68.2%, 15/22) of Enterobacteriaceae were ESBL producers; and were likely to be MDR bacteria compared to non-ESBL producers (aOR = 53.81, 95%CI 2.64–1095.73, p 0.01). (Table 2) Enterobacteriaceae displayed high rates of resistance to multiple antibiotics tested as presented in Table 3. *Pseudomonas aeruginosa* showed over 60% to 100% of resistance to commonly prescribed antibiotics including gentamicin (73%), ceftriaxone (100%), cefotaxime (100%), ceftazidime (70%) and ciprofloxacin (64%). (Table 3).

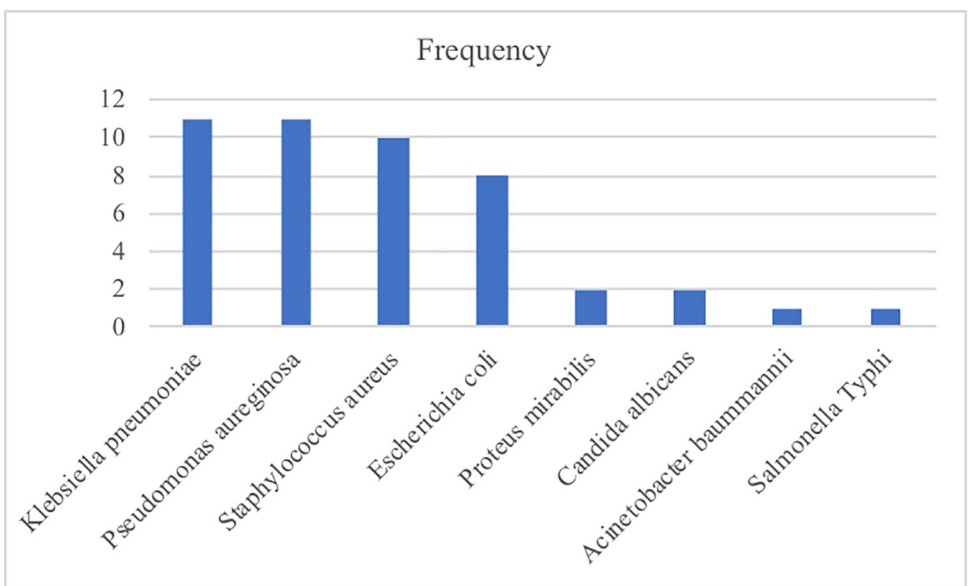

**Fig 1. Bacteria isolated from patients with BSI at MNH.**

**Table 2. Factors for isolation of multidrug resistant bacteria in confirmed Bloodstream infection.**

| Variable | Frequency | MDR %(n) | cOR | 95%CI | p value | aOR | 95%CI | p value |
|---|---|---|---|---|---|---|---|---|
| **Enterobacteriaceae ESBL status** | | | | | | | | |
| No | 7 | 14.3(1) | 1 | | | | | |
| Yes | 15 | 93.3(14) | 84 | 4.48–51576 | 0.003* | 53.81 | 2.64–1095.73 | 0.01* |
| **Antibiotic use before culture** | | | | | | | | |
| No | 26 | 53.8(14) | 1 | | | 1 | | |
| Yes | 20 | 85(17) | 4.57 | 1.14–20.70 | 0.03* | 2.88 | 0.53–11.53 | 0.2 |
| **ESBL** | | | | | | | | |
| No | 31 | 54.8(17) | 1 | | | 1 | | |
| Yes | 15 | 93.3(14) | 11.53 | 1.34–98.83 | 0.03* | 56.99 | 2.88–1128.44 | 0.008* |
| **Organism type** | | | | | | | | |
| GPC | 12 | 66.7(8) | 1 | | | | | |
| GNR | 34 | 67.6(23) | 1.05 | 0.26–4.24 | | | | |
| **Bacteria type** | | | | | | | | |
| Non-Enterobacteriaceae | 24 | 66.7 (16) | 1 | | | | | |
| Enterobacteriaceae | 22 | 68.2(15) | 1.07 | 0.31–3.68 | | | | |

cOR = crude odds ratio, aOR = adjustable odd ration, 95%CI = 95% Confidence interval

* significant

Remarkably, 46% of *Pseudomonas aeruginosa* were resistant to meropenem; furthermore, 27% (3/11) of them were carbapenemase producers. Resistance to piperacillin-tazobactam uncommonly used antibiotics was observed in 25% of *E. coli*, 20% of *Klebsiella pneumoniae* as well as 46% of *Pseudomonas aeruginosa*. *Staphylococcus aureus* displayed high resistance to ciprofloxacin (70%), Penicillin (80%) and erythromycin (70%); besides, (40%) of *Staphylococcus aureus* was MRSA. *Proteus mirabilis* and *Acinetobacter baumannii* were 100% resistant to all antibiotics tested.

## Predictors of mortality in bloodstream infection

In univariate analysis, patients with positive blood culture were two times more likely to die compared to those with negative blood culture (cOR 2.39, 95%CI 1.25–4.60, $p$ = 0.009). Patients aged more than 14 years suspected of BSI were at increased risk of mortality compared

**Table 3. Antimicrobial resistance pattern of bacteria isolated from patients with BSI at MNH.**

| Bacterial species | N | Percentage resistance | | | | | | | | | | | | | | |
|---|---|---|---|---|---|---|---|---|---|---|---|---|---|---|---|---|
| | | CN | CRO | CAZ | CTX | AMC | AK | MEM | PRT | CIP | SXT | FOX | DA | P | DOX | E |
| *E. coli* | 8 | 25 | 63 | 57 | 88 | 71 | - | - | 25 | 50 | 50 | NA | NA | NA | NA | NA |
| *K. pneumoniae* | 11 | 55 | 89 | 90 | 80 | 91 | 9 | - | 20 | 36 | 80 | NA | NA | NA | NA | NA |
| *P. mirabilis* | 2 | 100 | 100 | 100 | 100 | 100 | 100 | 100 | - | 100 | 100 | NA | NA | NA | NA | NA |
| *P. aeruginosa* | 11 | 73 | 100 | 70 | 100 | - | 55 | 46 | 46 | 64 | - | NA | NA | NA | NA | NA |
| *S. aureus* | 10 | 44 | NA | NA | NA | NA | NA | NA | NA | 70 | 10 | 40 | 20 | 80 | 60 | 70 |
| *A. baumannii* | 1 | - | 100 | 100 | 100 | 100 | - | 100 | 100 | 100 | - | NA | NA | NA | NA | NA |
| S. Typhi | 1 | 100 | - | - | - | - | - | - | - | - | - | NA | NA | NA | NA | NA |

Key: CN gentamicin, CRO ceftriaxone, CAZ ceftazidime, CTX cefotaxime, AMC amoxicillin-clavulanic acid, AK amikacin, MEM meropenem, PRT piperacillin-tazobactam, CIP ciprofloxacin, SXT trimethoprim-sulfamethoxazole, FOX cefoxitin DA clindamycin, P penicillin, DOX doxycycline, E erythromycin, NA Not Applicable

to those below (cOR 2.26, 95% 1.38–3.69, $p$ = 0.001). Patients suspected of BSI admitted to the surgical ward (cOR 2.61, 95%CI 1.32–5.18, $p$ = 0.006) and ICU (cOR 2.57, 95%CI 2.57–9.87, $p < 0.001$) were at increased risk of mortality compared to those admitted to the medical ward. Patients admitted for stroke/post-surgery/Diabetes mellitus were at higher risk of death (cOR 2.26, 95%CI 1.08–4.72, $p$ = 0.03) compared to those admitted with other underlying diseases. BSI with MDR bacteria were found to predict mortality (cOR 6.1, 95%CI 1.2–31.6, $p$ = 0.03) compared to non-MDR infection (Table 4).

Applying multivariate analysis, the result of blood culture, location of admission and status MDR was found to be independent risk factors for mortality in patients suspected of BSI. Patients with positive bacteria blood culture were twice as likely to die versus blood culture-negative patients (aOR 2.24, 95%CI 1.12–4.47, p 0.02). Admission to the ICU remained an independent risk factor for mortality with 4 times the risk of mortality compared to admission to the pediatric wards (aOR 3.88, 95%CI 1.60–9.41, p = 0.003). The presence of MDR bacteria was found to be an independent predictor for mortality with 16 times odds of dying compared to non-MDR bacterial infection (aOR 15.62, 95%CI 1.24–161.38, p = 0.02) (Table 4).

## Discussion

In this study conducted at a tertiary hospital setting, we found a slightly lower (11.4%) prevalence of BSI compared to previous findings at the same facility [1, 12]. However, our study observed a significantly high case fatality rate of 37%. Reports from various study populations have reported different prevalence rates of BSI [1, 4, 5, 12–16]. Blomberg et al. found a prevalence of 13.9% among admitted children at the same hospital [1]. Likewise, Moyo et al. reported a prevalence of 13.4% among all patients of different age groups at the same hospital [12]. Moyo et al included coagulase-negative *Staphylococcus* as the true pathogens in the analysis, which were not considered in our study. One important limitation of our study was the inclusion of patients that had used antibiotics before blood culture, this might explain the low rate of BSIs observed. A fraction of the negative cultures might have been false negatives. Also, we did not include anaerobic culture this might have excluded anaerobic bacteria, although these usually are low in number, compared to the aerobes. We propose in designing studies on BSI, efforts should be made to include catchment settings, where patients might not be exposed to antibiotics and include anaerobic culture. Another important caveat was the short duration of our study, which could have affected the prevalence observed. The prevalence of BSI may vary to geographical location, seasonality and population studied [5, 17, 18]. However, we tried to include all blood cultures processed during the study period and we had a reasonable sample size. Further research should be done over long time periods and different study populations to address other confounding factors.

As expected, patients admitted in ICU suspected with BSI had a higher risk of mortality compared to those admitted in non-ICU wards. This finding is in line with other earlier studies on risk factors for BSI [6, 8]. We observed that unlike patients admitted in non-ICU, patients in ICU were likely to have more severe underlying diseases. This observation could explain the observed high mortality. In these circumstances, our findings suggest prompt investigation in suspected BSI in ICU and appropriate antibiotic use guided by laboratory results. Previous studies have also highlighted the positive impact of prompt blood culture [4, 19].

Our study observed that positive bacterial blood culture was an independent laboratory predictor for mortality in patients suspected of BSI. A similar finding was also observed in previous studies from developing countries in the neonate [20], children [13], and adults [2]. This finding emphasizes the importance of blood culture in suspected BSI. It also highlights the role

**Table 4. Predictors of Mortality in Patients with suspected and confirmed Bloodstream infection.**

| Variable | Frequency | Death %(n) | cOR | 95%CI | p value | aOR | 95%CI | p value |
|---|---|---|---|---|---|---|---|---|
| **All patients with suspected BSI (n = 402)** | | | | | | | | |
| **Blood culture** | | | | | | | | |
| Negative | 356 | 19.7(70) | 1 | | | 1 | | |
| Positive | 46 | 37(17) | 2.39 | 1.25–4.60 | 0.009* | 2.24 | 1.12–4.47 | 0.02* |
| **Sex** | | | | | | | | |
| Female | 167 | 19.9(33) | 1 | | | | | |
| Male | 235 | 23(54) | 1.2 | 0.74–1.97 | 0.44 | | | |
| **Age (years)** | | | | | | | | |
| <14 | 211 | 15.2(332 | 1 | | | 1 | | |
| >14 | 191 | 28.8(55) | 2.26 | 1.38–3.69 | 0.001* | 01.22 | 0.59–2.53 | 0.6 |
| **Ward** | | | | | | | | |
| Pediatric | 164 | 12.2(20) | 1 | | | 1 | | |
| Surgical | 79 | 26.6(21) | 2.61 | 1.32–5.18 | 0.006* | 2.23 | 0.87–5.68 | 0.09 |
| Medical | 91 | 19.8(18) | 1.78 | 0.89–3.56 | 0.11 | 1.44 | 0.6–3.55 | 0.4 |
| ICU | 68 | 41.2(28) | 5.04 | 2.57–9.87 | <0.001* | 3.88 | 1.60–9.41 | 0.003* |
| **Underlying diseases** | | | | | | | | |
| Malignant | 93 | 16.1 (15) | 1 | | | 1 | | |
| Kidney diseases | 55 | 23.6 (13) | 1.61 | 0.70–3.70 | 0.3 | 0.85 | 0.34–2.12 | 0.7 |
| Sepsis | 131 | 18.3 (24) | 1.17 | 0.58–2.37 | 0.7 | 0.92 | 043–1.93 | 0.8 |
| Stroke/post-surgery/Diabetes mellitus | 76 | 30.3 (23) | 2.26 | 1.08–4.72 | 0.03* | 1.3 | 0.58–2.92 | 0.5 |
| Others | 47 | 25.5 (12) | 1.78 | 0.76–4.20 | 0.2 | 1.06 | 0.42–2.73 | 0.9 |
| **Patients with BSI (n = 46)** | | | | | | | | |
| **MDR** | | | | | | | | |
| No | 15 | 13.7(2) | 1 | | | 1 | | |
| Yes | 31 | 48.4(15) | 6.1 | 1.2–31.6 | 0.03* | 15.62 | 1.24–161.38 | 0.02* |
| **ESBL** | | | | | | | | |
| No | 31 | 35.5(11) | 1 | | | | | |
| Yes | 15 | 40(6) | 1.2 | 0.34–4.31 | 0.7 | | | |
| **Organism type** | | | | | | | | |
| GNR | 34 | 35.3(12) | 1 | | | | | |
| GPC | 12 | 41.7(5) | 1.31 | 0.34–5.03 | 0.7 | | | |
| **Bacteria type** | | | | | | | | |
| Enterobacteriaceae | 22 | 27.3(6) | | | | | | |
| Non-Enterobacteriaceae | 24 | 45.8(11) | 2.26 | 0.66–7.76 | 0.2 | | | |
| **Underlying diseases** | | | | | | | | |
| Malignant | 6 | 33.3 (2) | 1 | | | | | |
| Kidney diseases | 10 | 30 (3) | 0.86 | 0.98–7.5 | 0.9 | 0.15 | 0.01–3.05 | 0.2 |
| Sepsis | 15 | 33.3 (5) | 1 | 0.13–7.45 | 1 | 0.12 | 0.01–2.25 | 0.2 |
| Stroke/post-surgery/Diabetes mellitus | 8 | 62.5 (5) | 3.33 | 0.36–30.70 | 0.3 | 0.71 | 0.04–13.87 | 0.9 |
| Others | 7 | 28.6(2) | 0.8 | 0.076–8.47 | 0.9 | 0.27 | 0.01–5.74 | 0.4 |

cOR = crude odds ratio, aOR = adjustable odd ration, 95%CI = 95% Confidence interval, * significant

of the laboratory in prompt notifying clinicians when blood culture is positive as it could guide early choices of empiric antibiotics for better treatment outcomes. Besides, when the pathogen is identified and susceptibility results are available, the clinician needs to be alerted for them to adjust/de-escalation dosages of antibiotics. With the overall BSI case fatality rate of 37%,

further study needs to be performed to evaluate the impact of laboratory prompt notification of results.

Infection due to MDR bacteria was an independent predictor of mortality in our study. This finding was comparable to previous studies in Africa and elsewhere [2, 18]. Treatment of these infections is very difficult and carries poor prognosis as bacteria are resistant to all available antibiotics' options. Patients infected with MDR pathogens were supposed to receive reserve antibiotics like vancomycin, carbapenems or colistin; unfortunately, they are expensive and unavailable in our settings. Although performing and reporting phenotypic AST and defining MDR pathogens remain crucial for patients care, it is often not done in our settings. Our finding justifies the need for a clinical microbiology laboratory to notify and discuss with clinicians whenever they isolate MDR bacterium. Also, clinical microbiologists and infectious disease specialists need to work hand in hand in the management of MDR infected patients.

The current study observed that bacteria causing BSI were highly resistant to most antibiotics commonly used in our setting. The trend of resistance was comparable to resistance patterns observed in previous studies in the same setting [1, 12] and other hospital-associated infections [21]. Persistent high rates of resistance at our setting could be accounted for by increasingly empirical use of antibiotics; in most cases imprudent use of antibiotics. As observed in our study, antibiotic use before blood culture was an increased risk for isolation of MDR bacteria, similar to previous literature [22]. However, one weakness of this study is its restriction on current hospital use of antibiotics to assess risk factors for MDR bacteria. Recent publications highlight colonization by resistant bacteria due to antibiotic exposure prior to hospitalization; increase the risk of MDR infections [23–25]. We suggest that evaluation/audit of antibiotic use outside of health facilities (including illegal acquisition of antibiotics without a prescription) is also needed, to determine risk factors for future drug-resistant infections. The finding high rates of resistant bacteria causing BSI in Tanzania provide an opportunity for revising current practice on the management of BSI. Lately, the clinical microbiology laboratory does not produce an annual AMR report, to guide the clinician in an empiric antibiotic prescription. We strongly recommend the clinical microbiology laboratory to regularly produce local AMR reports.

The study revealed 68.2% of Enterobacteriaceae were ESBL producers. Blomberg et al. reported high rates of genotypic confirmed ESBL producing Enterobacteriaceae in children with BSI at the same hospital that predicted mortality [3]. Similarly, high rates of ESBL producing pathogens have been reported at the same hospital from clinical isolates of urine [26] and surgical site infections [21]. Beside ESBL, we also observed MRSA and phenotypic carbapenemase production in *Pseudomonas aeruginosa* and *Proteus mirabilis*. Carbapenemase-producing *Pseudomonas aeruginosa* has also been reported previously at our hospital [27]. It is worrying to note the growing resistance to carbapenems, which is the last effective therapy for severe Gram-negative infections. As observed in earlier studies, ESBL, MRSA, and carbapenemase-producing isolates carry a very poor prognosis and associated with increased health care costs [3, 20]. The findings warrant the need for a clinical microbiology laboratory to enforce the policy of detecting and reporting resistant pathogens. If enforced, it will help in the early detection of an outbreak due to these pathogens.

## Conclusion

The overall prevalence of BSI was 11.4%, and 17 of patients died (case fatality rate was 37%). The majority of the bacteria isolated from BSI were MDR. Admission to the ICU and positive blood culture were independently associated with mortality in suspected BSI. MDR bacteria were an independent predictor for mortality in confirmed BSI.

## Supporting information

**S1 Dataset.**
(XLS)

## Acknowledgments

The authors would like to thank the management of Muhimbili National Hospital for the support on reagents, supplies and other consumables that were used in the study.

## Author Contributions

**Conceptualization:** Joel Manyahi.

**Data curation:** Edna Mgimba.

**Formal analysis:** Joel Manyahi.

**Investigation:** Edna Mgimba.

**Writing – original draft:** Joel Manyahi.

**Writing – review & editing:** Upendo Kibwana, Mtebe Majigo.

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
