## [Decision Letter · Decision Letter 0]

26 Sep 2019

PONE-D-19-19558

Multi-drug resistance bacteria predict mortality in blood stream infection in a tertiary setting in Tanzania

PLOS ONE

Dear Dr Manyahi,

Thank you for submitting your manuscript to PLOS ONE. After careful consideration, we feel that it has merit but does not fully meet PLOS ONE’s publication criteria as it currently stands. Therefore, we invite you to submit a revised version of the manuscript that addresses the points raised during the review process.

Overall both reviews as well as myself find that  the manuscript contributes important clinical information to the scientific community on blood stream infections and multi-drug resistance.  However, both reviewers point out several areas which need to be clearly and sufficiently addressed prior to acceptance of the manuscript for publication. Please address both reviewers comments point by point.  The manuscript needs to be professionally edited for language by a copy editor as PLOS ONE does not provide such a service, as the manuscript stands there is a profound lack of clarity as detailed by both reviewers.  The manuscript will have a greater impact once the language and writing style are improved. In addition, the use of common clinical microbiology nomenclature in manuscripts is not consistent throughout the manuscript and could be revised, for example hyphenating at times but not others. As raised by reviewer #1 there are areas of data detailed within the results which were not addressed within the discussion or conclusions this needs to be resolved. There is also discussion that the subjects were receiving antibiotics (or at least some were).  The authors need to include this very important data into their data tables and analysis.  Specify which antibiotics were being administered and their duration of treatment then analyze to see if there is a direct correlation between the mdr found and the pre-treatment instead of just suggesting this to be a possibility.  The authors also need to include precise cause of death for all subjects, both the positive and the negative blood culture results.  Simply because the blood culture grew that may not have been the underlying cause death. Perhaps there was trauma which precipitated mortality and the bsi was simply secondary to the trauma.

We would appreciate receiving your revised manuscript by Nov 10 2019 11:59PM. To enhance the reproducibility of your results, we recommend that if applicable you deposit your laboratory protocols in protocols.io, where a protocol can be assigned its own identifier (DOI) such that it can be cited independently in the future. For instructions see: http://journals.plos.org/plosone/s/submission-guidelines#loc-laboratory-protocols

We look forward to receiving your revised manuscript.

Kind regards,

Anne Wertheimer, PhD

Academic Editor

PLOS ONE

Journal Requirements:

2. In ethics statement in the manuscript and in the online submission form, please provide additional information about the patient records/samples used in your retrospective study. Specifically, please ensure that you have discussed whether all data/samples were fully anonymized before you accessed them and/or whether the IRB or ethics committee waived the requirement for informed consent. If patients provided informed written consent to have data/samples from their medical records used in research, please include this information.

3. Our internal editors have looked over your manuscript and determined that it could be within the scope of our Antimicrobial Resistance call for papers. This collection of papers is headed by a team of Guest Editors for PLOS ONE: Kathryn Holt (Monash University and London School of Hygiene and Tropical Medicine), Alison H. Holmes (Imperial College London), Alessandro Cassini (WHO Infection Prevention and Control Global Unit), Jaap A. Wagenaar (Utrecht University). The Collection will encompass a diverse range of research articles; additional information can be found on our announcement page: https://collections.plos.org/s/antimicrobial-resistance. If you would like your manuscript to be considered for this collection, please let us know in your cover letter and we will ensure that your paper is treated as if you were responding to this call. If you would prefer to remove your manuscript from collection consideration, please specify this in the cover letter.

Reviewers' comments:

Reviewer's Responses to Questions

**Comments to the Author**

1. Is the manuscript technically sound, and do the data support the conclusions?

Reviewer #1: Yes

Reviewer #2: Yes

2. Has the statistical analysis been performed appropriately and rigorously? 

Reviewer #1: Yes

Reviewer #2: Yes

3. Have the authors made all data underlying the findings in their manuscript fully available?

Reviewer #1: Yes

Reviewer #2: No

4. Is the manuscript presented in an intelligible fashion and written in standard English?

Reviewer #1: No

Reviewer #2: Yes

5. Review Comments to the Author

Reviewer #1: The manuscript describes a cross-sectional retrospective analysis of bloodstream infections in a limited resource setting with the aim of examining the predictors of mortality of patients with BSI. As the authors indicate, there have been previous similar studies performed at the same setting therefore this work is not particularly original. However, given the limited data available on clinical outcomes of patients with BSI in limited resource settings the study is noteworthy to the scientific community.

In general, the publication could benefit from revision of the language to improve the grammar and readability. I suggest the authors work with a copyeditor, as this would be beneficial to the readers. In addition, the use of common clinical microbiology nomenclature in manuscripts is not consistent throughout the manuscript and could be revised, for example hyphenating at times but not others.

Methods:

Manufacturer name of branded reagents ie, Staphaurex, Strep grouping kit etc. should be included.

If a software application was used to perform the statistical analysis it should be included.

Results:

The Abstract references Proteus mirabilis, however results for P. mirabilis and A. baumannii are both excluded from any statements in the Results section even though they show high levels of resistance to the antimicrobials in Table 2 and may have had an effect on the mortality metrics. Whether their impact on BSI and mortality is unlikely due to a low N is unclear as they are not discussed in the text.

In addition, whether any trends exist by individual bacterial species is not clear given that the organisms are aggregated by type GNR or GPC. The publication would benefit from an analysis to sub stratify by bacteria or at minimum Enterobacteriaceae vs. non-Enterobacteriaceae. If this analysis was performed and there was no significant difference between the groups this should be indicated in the manuscript.

Figure 1 includes Salmonella typhi which is also never addressed in the results.

Consider condensing Table 1 and placing the percentage in parentheses after the frequency value for ease of reading.

Discussion

Though the authors indicate some limitations of the study, the impact of such limitations on the data and conclusions is not thoroughly critiqued.

The study was performed in April-May 2018. Given the short duration, the authors should address whether there is any seasonality that could may have affected prevalence or any other associated limitations.

Lines 255-264: the authors highlight the impact of antimicrobial stewardship benefits for BSI treatment however do not cite any references. There are ample publications on Antimicrobial stewardship and BSIs that should be referenced.

Reviewer #2: 1. The manuscript is written mostly in clear, standard English, but need some revision with regards to the use of language, sentence constructs, exclusion of words like "the" where it is actually required.

2. Some sentences could be revised to prevent ambiguity (e.g. Line 214: Patients aged more than 14 years suspected of BSI were.......").

3. Capitalization of some words needs to be corrected (Carbapenemases in middle of a sentence should be carbapenemases, "figure 1: in line 176 should be Figure 1, etc.). Another example is "Sixty" in line 180.

4. Line 215: the authors indicated that certain patients were at high risk of dying compared to those admitted at medical ward. Is that the intended word or did they mean "higher/increased risk"?

5. Line 183: Remarkably, 46% of Pseudomonas aeruginosa were resistance to meropenem; furthermore, 27% (3/11) of them were Carbapenemase producing pathogens.

“resistance” should be “resistant”; ‘carbapenemase” instead of “Carbapenemase”.

6. Table 2 needs some formatting, for example it might be better if the abbreviations of the antibiotics are in one row (not one letter of an abbreviation in a next row).

“Bacteria species” (in row 1 of the table) should be “Bacterial species”

Consistency is needed: all bacteria mentioned in Table 2 denoted with one letter abbreviation (e.g. S.aureus), except for Proteus mirabilis. As this organism were mentioned earlier in manuscript, please change to P.mirabilis, to be consistent with rest of table.

Correction: “Percent of bacteria resisted” – suggested change: “Percentage resistance”.

7. Table 3: The number of specimens included in study, based on the suspicion of BSI, equals 402. Yet the number of males and females in Table 3= 352 (117, 235) - should 117 be 167 perhaps?

According to Table3, 46 patients had culture-confirmed BSI, and in line173 it is stated that 46 pathogens were isolated. Yet 48 organisms were isolated (36 GNB,12 GPC) according to Table 3. If this is not an error, I assume one or two patients had a polymicrobial infection. If this assumption is correctL did the patient(s) with more than one organism also succumb to the infection(s), and which combination of organisms was isolated? The number of cases studied here obviously is little, but as our knowledge of polymicrobial infections grow (and the impact on patient outcome), this type of information is valuable. If the number was not a typo error, perhaps mention the specific organisms and patient outcome.

8. The authors discussed that "the hospital need to intensify infection prevention and control program to limit the spread of BSI and its associated mortality". While this statement is true for health care facilities in general, this specific paper did not look at any IPC practices and only briefly mentioned that all patients in this study received antibiotics prior to sampling. No IPC practices (or observed lack) were reported here, neither were studies done to determine if organisms, such as the ESBLs, were identical and spreading in the same facility. Perhaps revise the sentence to indicate that (1) failure of IPC COULD be one contributing factor in the spread of BSI infections in the specific setting, and (2) local IPC practices must be scrutinized to determine which areas needs to be implemented or improved.

Secondly, the authors mentioned that "Persistent of high rates of resistance at our setting could be accounted by increasingly empiric use of antibiotics; in most cases imprudent use of antibiotics.". Again, while this is most probably true, we did not see the data from this healthcare facility (not reported in this study) to substantiate this. The alternative should also be considered, that persons could be colonised by resistant bacteria due to antibiotic exposure prior to hospitalisation: Increasingly, we see publications of data indicating that outpatient management of UTIs (for example) also increase the risk of ESBL infections. This aspect (antibiotic use outside of health facilities) is neglected in many studies, though much needed especially in settings where antibiotic stewardship is neglected. Perhaps the authors could include a sentence to suggest that evaluation/audit of antibiotic use outside of health facilities (including illegal acquisition of antibiotics without prescription) is also needed, to determine risk factors for future drug resistant infections.

9. Regarding the culture negative BSIs: the authors noted that all samples were collected from patients already on antibiotics ("All patients included in the study had used antibiotics prior to blood culture, which might explain the observed low rate of BSI"). This is problematic, as this affects the recovery of organisms. While this might explain the low rate of BSIs, it is reasonable to assume that a fraction of the negative cultures might in fact have been false negatives. Another group of organisms also excluded by this study is anaerobic bacteria (although these usually are low in number, compared to the aerobes).

Summary:

The manuscript provides valuable information regarding aerobic organisms cultured from BSIs in a Tanzanian clinical setting. As noted by the authors, such information is limited and much needed for the relevant country. It also adds to the knowledge of drug resistant bacteria on the continent. Of concern is the high number of MDR isolates reported here, particularly the ESBLs. While the study (unfortunately) were done with samples taken after antibiotic therapy was initiated, the collected data still is valuable, albeit potentially skewed towards drug resistant organisms.

Thus, while the manuscript needs some language revisions and attention must be given to some comments, it presents us with much needed data, presented in (mostly) clear English language. This manuscript is therefore recommended for publication, with revisions as suggested.

6. PLOS authors have the option to publish the peer review history of their article (what does this mean?). If published, this will include your full peer review and any attached files.

Reviewer #1: No

Reviewer #2: No

---

## [Author Response · Author response to Decision Letter 0]

26 Nov 2019

Response to editor

As raised by reviewer #1 there are areas of data detailed within the results which were not addressed within the discussion or conclusions this needs to be resolved. There is also discussion that the subjects were receiving antibiotics (or at least some were). The authors need to include this very important data into their data tables and analysis. Specify which antibiotics were being administered and their duration of treatment then analyze to see if there is a direct correlation between the mdr found and the pre-treatment instead of just suggesting this to be a possibility. The authors also need to include precise cause of death for all subjects, both the positive and the negative blood culture results. Simply because the blood culture grew that may not have been the underlying cause death. Perhaps there was trauma which precipitated mortality and the bsi was simply secondary to the trauma.

Response

Thank you for your valuable comments to improve our manuscript. We have tried to discuss all key findings in our discussion as suggested. All comment on important results missing narrative in text, these has been addressed accordingly.

History of antibiotic use has been added on table 1 and new table 2. It was difficult to get information on exact duration of treatment of these antibiotics, but analysis has been done looking at correlation of antibiotic use prior to culture and isolation of multi-drug resistant bacteria (Table 2) and explained in result section.We have also included analysis of underlying diseases on prediction of mortality on table 4.

Response to reviewer 

Reviewer #1

Comment

The manuscript describes a cross-sectional retrospective analysis of bloodstream infections in a limited resource setting with the aim of examining the predictors of mortality of patients with BSI. As the authors indicate, there have been previous similar studies performed at the same setting therefore this work is not particularly original. However, given the limited data available on clinical outcomes of patients with BSI in limited resource settings the study is noteworthy to the scientific community. In general, the publication could benefit from revision of the language to improve the grammar and readability. I suggest the authors work with a copyeditor, as this would be beneficial to the readers. In addition, the use of common clinical microbiology nomenclature in manuscripts is not consistent throughout the manuscript and could be revised, for example hyphenating at times but not others.

Response

We appreciate for a comment, and we have addressed microbiology nomenclature thought the manuscript and hyphenating has been addressed as well. study. 

Comment

Methods: Manufacturer name of branded reagents ie, Staphaurex, Strep grouping kit etc. should be included. If a software application was used to perform the statistical analysis it should be included.

Response:

Thank you for the comment. We have included the manufacturers of the reagents as suggested line 104 and 105. Also, the software application used has been added, line 128.

Comment

The Abstract references Proteus mirabilis, however results for P. mirabilis and A. baumannii are both excluded from any statements in the Results section even though they show high levels of resistance to the antimicrobials in Table 2 and may have had an effect on the mortality metrics. Whether their impact on BSI and mortality is unlikely due to a low N is unclear as they are not discussed in the text.

Response

We thank you for this valuable comment, we have added narrative in our text on resistant pattern for Proteus mirabilis and Acinetobacter baummannii, line 192 – 193.

Comment

In addition, whether any trends exist by individual bacterial species is not clear given that the organisms are aggregated by type GNR or GPC. The publication would benefit from an analysis to sub stratify by bacteria or at minimum Enterobacteriaceae vs. non-Enterobacteriaceae. If this analysis was performed and there was no significant difference between the groups this should be indicated in the manuscript.

Response

Due to small number of individual bacteria stratifications did not bring any significant differences. However, we sub stratified to Enterobacteriaceae and non-Enterobacteriaceae. Significant finding was observed on univariate analysis for isolation of multi-drug resistant bacteria Table 2 and line 183 – 185, but not on mortality table 4.

Comment

Figure 1 includes Salmonella typhi which is also never addressed in the results.

Response

Salmonella Typhi has been included in result line 174 and on table 3.

Comment

Consider condensing Table 1 and placing the percentage in parentheses after the frequency value for ease of reading

Response

Suggestion accepted and included in table 1

Comment

Though the authors indicate some limitations of the study, the impact of such limitations on the data and conclusions is not thoroughly critiqued

Response

We have highlighted impact of our limitation on data interpretation and conclusion, line 261 – 272

Comment

The study was performed in April-May 2018. Given the short duration, the authors should address whether there is any seasonality that could may have affected prevalence or any other associated limitations.

Response

Thank you for comment Line, we have highlighted this in our discussion line 268 - 272

Comment

Lines 255-264: the authors highlight the impact of antimicrobial stewardship benefits for BSI treatment however do not cite any references. There are ample publications on Antimicrobial stewardship and BSIs that should be referenced.

Response

Citations have been added, line 279 – 280.

Reviewer #2: 

Comment

1.The manuscript is written mostly in clear, standard English, but need some revision with regards to the use of language, sentence constructs, exclusion of words like "the" where it is actually required.

Response

Thank you, we have done our best. The manuscript was edited for language by a copy editor

Comment

2. Some sentences could be revised to prevent ambiguity (e.g. Line 214: Patients aged more than 14 years suspected of BSI were.......").

Response

Line 214 has been revised, now line 229 - 231

Comment

3. Capitalization of some words needs to be corrected (Carbapenemases in middle of a sentence should be carbapenemases, "figure 1: in line 176 should be Figure 1, etc.). Another example is "Sixty" in line 180.

Response

Capitalization of some words have been worked on thought the document. For example, Figure 1 as seen in line 176 and line 183 Sixty has been corrected

Comment

4. Line 215: the authors indicated that certain patients were at high risk of dying compared to those admitted at medical ward. Is that the intended word or did they mean "higher/increased risk"?

Response

Thank you, suggestion has been accepted we meant higher/increased risk, now line 231 – 233.

Comment

5. Line 183: Remarkably, 46% of Pseudomonas aeruginosa were resistance to meropenem; furthermore, 27% (3/11) of them were Carbapenemase producing pathogens.

“resistance” should be “resistant”; ‘carbapenemase” instead of “Carbapenemase”.

Response

Correction have been made, Line 187– 189

Comment

6. Table 2 needs some formatting, for example it might be better if the abbreviations of the antibiotics are in one row (not one letter of an abbreviation in a next row).

“Bacteria species” (in row 1 of the table) should be “Bacterial species”

Consistency is needed: all bacteria mentioned in Table 2 denoted with one letter abbreviation (e.g. S. aureus), except for Proteus mirabilis. As this organism were mentioned earlier in manuscript, please change to P.mirabilis, to be consistent with rest of table.

Correction: “Percent of bacteria resisted” – suggested change: “Percentage resistance”.

Response

All suggestions have been accepted, now it is table 3.

Comment

7. Table 3: The number of specimens included in study, based on the suspicion of BSI, equals 402. Yet the number of males and females in Table 3= 352 (117, 235) - should 117 be 167 perhaps?

Response

Thank you for noting this, it was typo error, now it has been addressed. Table 4

Comment

According to Table3, 46 patients had culture-confirmed BSI, and in line173 it is stated that 46 pathogens were isolated. Yet 48 organisms were isolated (36 GNB,12 GPC) according to Table 3. If this is not an error, I assume one or two patients had a polymicrobial infection. If this assumption is correctly did the patient(s) with more than one organism also succumb to the infection(s), and which combination of organisms was isolated? The number of cases studied here obviously is little, but as our knowledge of polymicrobial infections grow (and the impact on patient outcome), this type of information is valuable. If the number was not a typo error, perhaps mention the specific organisms and patient outcome.

Response

There were no polymicrobial infection. This was an error and has been addressed on table 4.

Comment

8. The authors discussed that "the hospital need to intensify infection prevention and control program to limit the spread of BSI and its associated mortality". While this statement is true for health care facilities in general, this specific paper did not look at any IPC practices and only briefly mentioned that all patients in this study received antibiotics prior to sampling. No IPC practices (or observed lack) were reported here, neither were studies done to determine if organisms, such as the ESBLs, were identical and spreading in the same facility. Perhaps revise the sentence to indicate that (1) failure of IPC COULD be one contributing factor in the spread of BSI infections in the specific setting, and (2) local IPC practices must be scrutinized to determine which areas needs to be implemented or improved.

Response

Thank you for comment, we have omitted the sentence in our discussion.

Comment

Secondly, the authors mentioned that "Persistent of high rates of resistance at our setting could be accounted by increasingly empiric use of antibiotics; in most cases imprudent use of antibiotics.". Again, while this is most probably true, we did not see the data from this healthcare facility (not reported in this study) to substantiate this. The alternative should also be considered, that persons could be colonized by resistant bacteria due to antibiotic exposure prior to hospitalization: Increasingly, we see publications of data indicating that outpatient management of UTIs (for example) also increase the risk of ESBL infections. This aspect (antibiotic use outside of health facilities) is neglected in many studies, though much needed especially in settings where antibiotic stewardship is neglected. Perhaps the authors could include a sentence to suggest that evaluation/audit of antibiotic use outside of health facilities (including illegal acquisition of antibiotics without prescription) is also needed, to determine risk factors for future drug resistant infections.

Response

In table 1, we have included antibiotics prescribed to these patients before collection of blood. Also, to support our argument we have added table 2, were we found antibiotic use prior to culture had correlated to isolation of MDR bacteria on univariate analysis. In addition, all comments have been accepted line 311 – 315. 

Comment

9. Regarding the culture negative BSIs: the authors noted that all samples were collected from patients already on antibiotics ("All patients included in the study had used antibiotics prior to blood culture, which might explain the observed low rate of BSI"). This is problematic, as this affects the recovery of organisms. While this might explain the low rate of BSIs, it is reasonable to assume that a fraction of the negative cultures might in fact have been false negatives. Another group of organisms also excluded by this study is anaerobic bacteria (although these usually are low in number, compared to the aerobes).

Response

We appreciate for suggestions and have been included in discussion line 263 - 265

---

## [Decision Letter · Decision Letter 1]

24 Dec 2019

PONE-D-19-19558R1

Multi-drug resistance bacteria predict mortality in blood stream infection in a tertiary setting in Tanzania

PLOS ONE

Dear Dr Manyahi,

Thank you for submitting your manuscript to PLOS ONE. After careful consideration, we feel that it has merit but does not fully meet PLOS ONE’s publication criteria as it currently stands. Therefore, we invite you to submit a revised version of the manuscript that addresses the points raised during the review process.

We would appreciate receiving your revised manuscript by Feb 07 2020 11:59PM. To enhance the reproducibility of your results, we recommend that if applicable you deposit your laboratory protocols in protocols.io, where a protocol can be assigned its own identifier (DOI) such that it can be cited independently in the future. For instructions see: http://journals.plos.org/plosone/s/submission-guidelines#loc-laboratory-protocols

We look forward to receiving your revised manuscript.

Kind regards,

Anne Wertheimer, PhD

Academic Editor

PLOS ONE

Additional Editor Comments (if provided):

Both reviewers expressed that the authors have made significant improvements on the manuscript. However, they also both request several minor revisions to improve the overall quality of the manuscript. Upon completion of these final revisions the manuscript will be suitable for publication. Again, the manuscript provides important clinical information beneficial to both the basic and clinical scientific community.

Reviewers' comments:

Reviewer's Responses to Questions

**Comments to the Author**

1. If the authors have adequately addressed your comments raised in a previous round of review and you feel that this manuscript is now acceptable for publication, you may indicate that here to bypass the “Comments to the Author” section, enter your conflict of interest statement in the “Confidential to Editor” section, and submit your "Accept" recommendation.

Reviewer #1: (No Response)

Reviewer #2: All comments have been addressed

2. Is the manuscript technically sound, and do the data support the conclusions?

Reviewer #1: Yes

Reviewer #2: Yes

3. Has the statistical analysis been performed appropriately and rigorously? 

Reviewer #1: Yes

Reviewer #2: N/A

4. Have the authors made all data underlying the findings in their manuscript fully available?

Reviewer #1: Yes

Reviewer #2: Yes

5. Is the manuscript presented in an intelligible fashion and written in standard English?

Reviewer #1: No

Reviewer #2: Yes

6. Review Comments to the Author

Reviewer #1: The major revisions have been adequately addressed however minor revisions to the writing are still necessary. Several examples are outlined below of modifications that can improve the writing if applied to the entire manuscript.

Line 25, Line 51 and Table Titles…”Blood-stream” vs. “bloodstream” vs “Blood stream” should be modified to ensure consistent use of one version throughout the manuscript.

Also line 30 and 82 “in-patient” vs “inpatient” should be modified to ensure consistency.

Line 30 “all inpatients at THE clinical”

Line 39, 293, 337 “resistant” not “resistance”

Line 39 and Line 45…multi-drug resistant vs Multi-drug resistant… Can be capitalized on line 39 to highlight the use of an abbreviation (MDR) in the remainder of the manuscript. Also for consistency, MDR can be used in lieu of “multi-drug resistant” throughout the remainder of the manuscript once define, ie line 236, 244, 293 etc.

Line 41 “Carbapenemase production”…remove the “s”

Line 176 capitalize Candida and italicize both Candida and albicans

Line 189 Piperacillin should be lowercase

Line 190-193 after the first use of the full genus and species the genus can be abbreviated. For example, line 190 Escherichia coli can be abbreviated to E. coli throughout the remainder of the manuscript as it has been defined on line 122. All other organisms should be modified for consistency.

Line 174 and Table 3 typhi should be capitalized and not italicized

Line 235 “admitted WITH other underlying diseases”

Line 260 Coagulase should be lowercase

Varied use of “in” or “at” for admission to the wards. Suggest modifying to “to the”…for example Lines 242 and 274-275… “admitted to the ICU” would allow for easier reading. Several other examples exist in the manuscript that should be modified as such.

Figure 1. Be consistent with Genus and Species or Abbreviation and species for all organisms

Reviewer #2: The authors made various corrections and improvements to the manuscript. This paper will be a valuable contribution to our knowledge of BSIs, associated risk factors, etc. on the continent. It potentially could be used to improve on future study designs (especially to account for factors one has no control over (antibiotic use before hospitalisation, lack of stewardship in some settings etc).

Before final acceptance, some additional comments (minor really):

Two spelling errors still present: The word "malignant" is incorrectly spelled as “Malginant” in Tables 2 and 4.

The authors need to pay attention to the references, please modify the abbreviationsof journal names for the sake of uniformity (Some abbreviations are not written with the first letter as a capital letter etc). Correct Example: J Clin Gastroenterol. A number of references not abbreviated and not with same format. Examples of incorrect citation: South African medical

journal = Suid-Afrikaanse tydskrif vir geneeskunde; also "International journal of biological and medical research."

Other then these minimal issues stated above, congratulations to the authors for contributing valuable data lacking in many countries on the continent.

7. PLOS authors have the option to publish the peer review history of their article (what does this mean?). If published, this will include your full peer review and any attached files.

Reviewer #1: No

Reviewer #2: No

---

## [Author Response · Author response to Decision Letter 1]

28 Jan 2020

Response to reviewers

Reviewer #1

Comment

The major revisions have been adequately addressed however minor revisions to the writing are still necessary. Several examples are outlined below of modifications that can improve the writing if applied to the entire manuscript.

Response

We appreciate for a comment, and we have addressed all examples outlined below and other parts

Comment

Line 25, Line 51 and Table Titles “Blood-stream” vs. “bloodstream” vs “Blood stream” should be modified to ensure consistent use of one version throughout the manuscript.

Response

We have the comment and used one version `Bloodstream` throughout the manuscript.

Comment

Also line 30 and 82 “in-patient” vs “inpatient” should be modified to ensure consistency.

Response

We have addressed the comment and ensured consistency in using one version “inpatient” throughout the manuscript

Comment

Line 30 “all inpatients at THE clinical”

Response

The word the has been added in line 31

Comment

Line 39, 293, 337 “resistant” not “resistance”

Response

We have addressed the comment throughout the manuscript, for example line 40, 367 etc

Comment

Line 39 and Line 45…multi-drug resistant vs Multi-drug resistant… Can be capitalized on line 39 to highlight the use of an abbreviation (MDR) in the remainder of the manuscript. Also, for consistency, MDR can be used in lieu of “multi-drug resistant” throughout the remainder of the manuscript once define, ie line 236, 244, 293 etc.

Response

Thank you for comment, we have addressed the comment and abbreviated MDR on its first use. Also maintained consistency on use of MDR throughout the manuscript.

Comment

Line 41 “Carbapenemase production” …remove the “s”

Response

S has been removed as proposed line 42.

Comment

Line 176 capitalize Candida and italicize both Candida and albicans

Response

The comment has been addressed line 188

Comment

Line 189 Piperacillin should be lowercase

Response

The comment has been addresses in line 209

Comment

Line 190-193 after the first use of the full genus and species the genus can be abbreviated. For example, line 190 Escherichia coli can be abbreviated to E. coli throughout the remainder of the manuscript as it has been defined on line 122. All other organisms should be modified for consistency.

Response

Thank you for comment, we have managed to abbreviation for only Escherichia coli, but the rest we have maintained full names of genus and species. This could make easy readability of our work.

Comment

Line 174 and Table 3 typhi should be capitalized and not italicized

Response

Thank you, the comment has been addressed in line 186

Comment

Line 235 “admitted WITH other underlying diseases”

Response

With has been added in line 260.

Comment

Line 260 Coagulase should be lowercase

Response

The comment has been addressed in line 295

Comment

Varied use of “in” or “at” for admission to the wards. Suggest modifying to “to the”…for example Lines 242 and 274-275… “admitted to the ICU” would allow for easier reading. Several other examples exist in the manuscript that should be modified as such.

Response

We appreciate for the comment, modification has been made throughout the manuscript. For example, from line 256 to 269.

Comment

Figure 1. Be consistent with Genus and Species or Abbreviation and species for all organisms

Response

We have made modification and used the full name of genus and species

Reviewer #2: 

Comment

The authors made various corrections and improvements to the manuscript. This paper will be a valuable contribution to our knowledge of BSIs, associated risk factors, etc. on the continent. It potentially could be used to improve on future study designs (especially to account for factors one has no control over (antibiotic use before hospitalization, lack of stewardship in some settings etc).

Response 

Thank you for the comment and will look forward on designing other studies taking account for factors highlighted.

Comment

Two spelling errors still present: The word "malignant" is incorrectly spelled as “Malginant” in Tables 2 and 4.

Response

This has been addressed in table 1 and 4

Comment

The authors need to pay attention to the references, please modify the abbreviations of journal names for the sake of uniformity (Some abbreviations are not written with the first letter as a capital letter etc). Correct Example: J Clin Gastroenterol. A number of references not abbreviated and not with same format. Examples of incorrect citation: South African medical journal = Suid-Afrikaanse tydskrif vir geneeskunde; also "International journal of biological and medical research."

Response

Thank you, we have noted that and we written correct abbreviation throughout the references

---

## [Editor Report · Decision Letter 2]

13 Feb 2020

Multi-drug resistant bacteria predict mortality in blood stream infection in a tertiary setting in Tanzania

PONE-D-19-19558R2

Dear Dr. Manyahi,

We are pleased to inform you that your manuscript has been judged scientifically suitable for publication and will be formally accepted for publication once it complies with all outstanding technical requirements.

With kind regards,

Anne Wertheimer, PhD

Academic Editor

PLOS ONE
---

## [Editor Report · Acceptance letter]

21 Feb 2020

PONE-D-19-19558R2 

Multi-drug resistant bacteria predict mortality in bloodstream infection in a tertiary setting in Tanzania 

Dear Dr. Manyahi:

I am pleased to inform you that your manuscript has been deemed suitable for publication in PLOS ONE. Congratulations! Your manuscript is now with our production department. 

With kind regards,

on behalf of

Dr. Anne Wertheimer 

Academic Editor

PLOS ONE